# Wound Modulations in Glaucoma Surgery: A Systematic Review

**DOI:** 10.3390/bioengineering11050446

**Published:** 2024-04-30

**Authors:** Bhoomi Dave, Monica Patel, Sruthi Suresh, Mahija Ginjupalli, Arvind Surya, Mohannad Albdour, Karanjit S. Kooner

**Affiliations:** 1Department of Ophthalmology, University of Texas Southwestern Medical Center, Dallas, TX 75390, USA; bhd27@drexel.edu (B.D.); monica.patel2@utsouthwestern.edu (M.P.); sruthi.prabhasuresh@utsouthwestern.edu (S.S.); mahija.ginjupalli@utsouthwestern.edu (M.G.); arvindsurya52@gmail.com (A.S.); 2Drexel University College of Medicine, Philadelphia, PA 19129, USA; 3Department of Ophthalmology, King Hussein Medical Center Royal Medical Services, Amman 11180, Jordan; drmohannadalbdour@yahoo.com; 4Department of Ophthalmology, Veteran Affairs North Texas Health Care System Medical Center, Dallas, TX 75216, USA

**Keywords:** glaucoma surgery, wound healing, antifibrotic agents, anti-vascular endothelial growth factors, cytokine inhibitors, anti-LOXL2 monoclonal Ab, integrin inhibitors, growth factor inhibitors

## Abstract

Excessive fibrosis and resultant poor control of intraocular pressure (IOP) reduce the efficacy of glaucoma surgeries. Historically, corticosteroids and anti-fibrotic agents, such as mitomycin C (MMC) and 5-fluorouracil (5-FU), have been used to mitigate post-surgical fibrosis, but these have unpredictable outcomes. Therefore, there is a need to develop novel treatments which provide increased effectiveness and specificity. This review aims to provide insight into the pathophysiology behind wound healing in glaucoma surgery, as well as the current and promising future wound healing agents that are less toxic and may provide better IOP control.

## 1. Introduction

Glaucoma is the second leading cause of blindness, affecting more than 80 million patients worldwide and over 3 million in the USA [1]. The prevalence of glaucoma is expected to double over the next 30 years, which will pose a major public health challenge [2]. An important modifiable risk factor is elevated intraocular pressure (IOP) due to the blockage of aqueous humor (AH) outflow [3]. Therefore, it is imperative to understand the mechanics and dynamics of AH production and outflow. The drainage of AH occurs mainly through the conventional pathway [trabecular meshwork (TM), Schlemm’s canal (SC), collector channels, aqueous veins, and episcleral veins (EVs) 70%], as well as the non-conventional uveoscleral–uveovortex (US-UV) pathway, uveal meshwork, anterior face of the ciliary muscle through the muscle bundles, suprachoroidal space, and out through the sclera (30%), as in Figure 1A,B [4]. Though the dysfunction of the conventional pathway is not well understood, increased TM contractility, changes in extracellular matrix (ECM) composition, decreased pore density of the inner wall of SC, and disruption of local regulatory mediators may contribute to increased AH outflow resistance [5].

The initial conventional treatment options for controlling elevated IOP involve medications and laser procedures. If medications and laser treatment fail to lower IOP, the next step is performing incisional surgery, which includes trabeculectomy, trabeculotomy, glaucoma drainage devices (GDDs), and minimally invasive glaucoma surgeries (MIGS) [6,7,8]. Trabeculectomy is designed to remove a portion of the TM and SC to allow the flow of AH to the subconjunctival space [9]. Trabeculotomy, on the other hand, is performed either by an ab interno or ab externo approach, to perforate the TM. It is commonly used in children with congenital glaucoma, but rarely in adults. GDDs (Ahmed^®^, New World Medical Inc., Rancho Cucamonga, CA, USA; Baerveldt^®^, Advanced Medical Optics Inc., Santa Ana, CA, USA; and Molteno Ophthalmic Limited^®^, Dunedin, New Zealand), which consist of a tube and a plate, also drain the AH from the AC into the subconjunctival space, but with more controlled AH outflow and somewhat predictable outcomes. Recently, MIGS (Xen Gel Stent^®^ Abbvie/Allergan Co. Dublin, Ireland, PreserFlo^®^ MicroShunt [made of poly(styrene-b-isobutylene-b-styrene), or SIBS], Santen Pharmaceutical Company^®^, Osaka, Japan, iStent^®^ Glaukos Corp. inject, Hydrus^®^ Alcon microstents, Kahook^®^ New World Medical dual-blade goniotomy, trabectome^®^ MicroSurgical Technology, gonioscopy-assisted transluminal trabeculotomy, Glaucoma Associates of Texas, Trab 360 OMNI^®^ Sight Sciences, Inc., Visco 360 OMNI^®^ Sight Sciences, Inc., Ab interno canaloplasty (ABiC) Ellex Australia, and Streamline^®^ New World Medical Inc. surgical system have gained popularity due to their relatively quick insertion and lesser tissue manipulation [6,7,8]. A 2017 survey, conducted to assess surgical practice patterns among members of the American Glaucoma Society (AGS), showed a significant increase in the use of GDDs since 1997 [8]. Trabeculectomy remains the procedure of choice, with higher mean percentages of use (59% ± 30%), followed by GDDs (23% ± 23%) and MIGS (14% ± 20%) [8].

A significant postoperative complication of all incisional glaucoma surgeries is a vigorous fibroproliferative response leading to the blockage of AH outflow in the subconjunctival space (“ring-of-steel”), leading to inadequate control of IOP and surgical failure. Therefore, modulating the wound healing process is critical for optimal outcomes in the surgical management of glaucoma [10], hence the reason for this review.

## 2. Materials and Methods

### 2.1. Initial Search (Figure 2)

We followed the standards outlined by the Preferred Reporting Items for Systematic Reviews and Meta-Analyses (PRISMA) guidelines during data collection, and the PICOS (Population, Intervention, Comparison, Outcomes and Study) framework was used to create eligibility criteria, Table 1 [11]. We used keywords and MeSH terms, such as “glaucoma” (or “Glaucoma, Angle-Closure” or “Glaucoma, Open-Angle”), “glaucoma wound healing” or “glaucoma filtration surgery” (or “sclerostomy,” “trabeculectomy”, or “GDDs”), “anti-inflammatory agents” (or “antifibrotic agents”).

**Figure 2 bioengineering-11-00446-f002:**
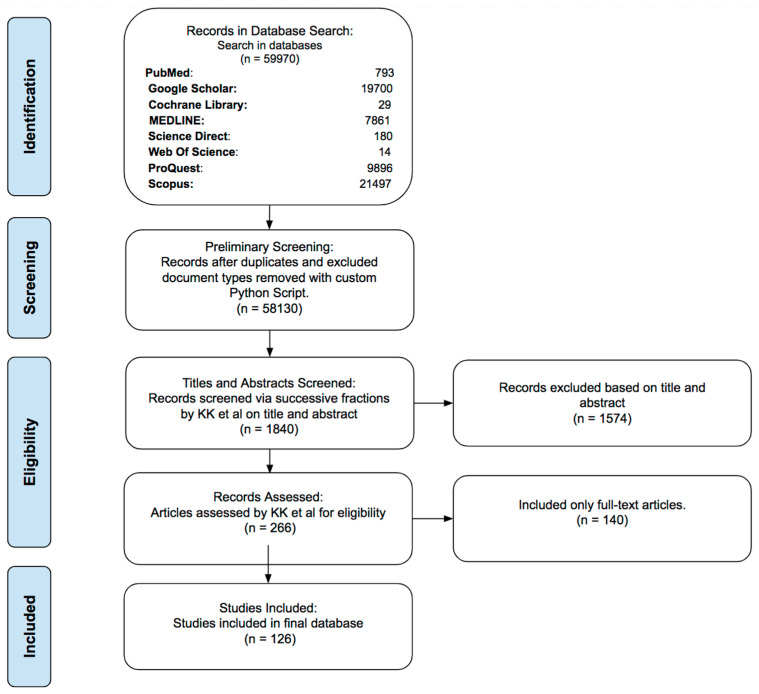
PRISMA Flowchart illustrating the selection process for this systematic review.

Using these terms, we systematically searched the online databases of PubMed (MEDLINE), Cochrane Library, ScienceDirect, Scopus, Google Scholar, ProQuest, and Web of Science up to June 2023. The records from the different databases were compiled in a comma-separated values (CSV) file on Google Sheets.

### 2.2. Preliminary Screening

We excluded non-English articles and study types, such as conference abstracts, commentaries, and duplicate papers with the same digital object identifier (DOI) using a script written in the Python programming language (Python Software Foundation, Wilmington, DE, USA, version 3.12.2). The selected manuscripts were then stored in the CSV for eligibility assessment and included information on authors, title, date of publication, journal, and DOI.

### 2.3. Eligibility Assessment

The four reviewers (KK, SS, MP, and BD) screened every article in the CSV for accuracy and best fit. We included full-text English articles, studies involving animal or human subjects, and clinical trials.

## 3. Results

After multiple rounds of screening, 126 studies were included in our review. The selected studies discussed the basic principles, development, and applications of wound healing modulation in glaucoma surgery. Figure 2 depicts the eligibility assessment process.

### 3.1. Overview of the Wound Healing Process

Understanding the conjunctival, episcleral, and scleral wound healing process is critical to evaluate wound healing modulation in glaucoma surgery. The wound repair process can be divided into four key phases: hemostasis, inflammation, proliferation, and remodeling (Figure 3) [12].

#### 3.1.1. Hemostasis

In the first stage, hemostasis, a platelet plug forms to prevent excessive blood loss. This is achieved through activation of the clotting cascades, which begins with vasoconstriction initiated by the release of thromboxane (TXA2) and endothelin-1 from the damaged endothelium [13]. Following this, the interaction of platelet receptors and ECM proteins (collagen, elastin, fibronectin) occurs to promote adherence to the walls of the surrounding blood vessels [14]. Once the platelet receptors adhere to the blood vessels, thrombin is promoted to activate platelets and release granules, which reinforce the coagulation process [15]. Concurrently, platelets produce platelet-derived growth factor (PDGF), which activates endothelial cells to repair damaged vasculature through angiogenesis. Once completed, the hemostasis phase is downregulated by inhibitors, such as activated protein C, prostacyclin, and antithrombin III [15].

#### 3.1.2. Inflammation

The second phase, inflammation, consists of the recruitment of immune cells designed to remove necrotic tissue and pathogens [10]. This phase of wound healing is initiated by the release of damage-associated molecular patterns (DAMP) molecules, pathogen-associated molecular patterns (PAMP) molecules, hydrogen peroxide (H_2_O_2_), lipid mediators, and chemokines from injured cells [16]. DAMPs are endogenous molecules consisting of DNA, peptides, ECM components, and ATP to activate the innate immune system, while PAMPs work to activate immune cells and release pro-inflammatory cytokines. Both patterns share a common goal of attracting leukocytes to the injured tissue. These modulators lead to the influx of immune cells, specifically neutrophils and leukocytes [17]. Once neutrophils are generated in the bone marrow, they are attracted to the site of injury by the “find me” signals from chemoattractants (molecules that promote movement), including DAMPs, H_2_O_2_, lipid mediators, and chemokines [17]. After traveling to the wound from damaged vessels, neutrophils remove necrotic tissue, and pathogens trap and kill pathogens with extracellular traps, resulting in wound decontamination [17]. Monocytes work by differentiating into macrophages with variable phenotypes, ultimately initiating the macrophage inflammatory response and further augmenting it by attracting additional monocytes [18].

#### 3.1.3. Proliferation

The proliferative phase is characterized by wound closure and is essential to wound healing. Proliferation may occur as early as 12 h post-injury, resulting in the formation of highly vascularized granulation tissue. This newly formed tissue allows for ECM synthesis and the activation of fibroblasts. This process occurs simultaneously with neovascularization and immunomodulation, contributing to wound contraction [16]. Wound contraction occurs when myofibroblasts grip the wound edges and pull them together [19]. Microvascular endothelial cells (ECs) lining blood vessels are central to neovascularization. Their activation relies on growth factors (a bioactive molecule released into the environment which affects cell growth) produced by nearby cells, and the production of proteolytic enzymes (matrix metalloproteinases (MMPs), disintegrins, and metalloproteinases) facilitates their navigation through the fibrin/fibronectin clot. ECs initiate angiogenesis by sprouting in response to pro-angiogenic signals (VEGF, FGF, PDGF-β, TGF-β) and angiopoietins, leading to proliferation and migration [15]. The new granulation tissue typically exhibits a red or pink color, attributed to the presence of new blood vessels and other inflammatory agents. The color and condition of the granulation tissue serve as indicators of the progress of wound healing. On the other hand, the dark granulation tissue is an evidence of poor perfusion, ischemia, or infection. This phase of wound healing can span from six days to up to three weeks or longer [14].

#### 3.1.4. Remodeling

In the final phase, remodeling, the granulation tissue is gradually replaced by normal connective tissue. This stage involves a decrease in tissue cellularity due to the massive apoptosis of fibroblasts, myofibroblasts, endothelial cells, and pericytes (cells that are embedded within the vessel wall endothelium) [20]. Integrins play a key role in facilitating cell attachment to the ECM. They have the ability to trigger the activation of latent transforming growth factor beta-1 (TGF-β1), which in turn regulates the processes of wound inflammation and the formation of granulation tissue [21]. The accumulation of ECM molecules, specifically collagen, is a hallmark of the remodeling phase. Type 3 collagen is converted to Type 1, which is a more mature and stiff form. This increases the tensile strength and elasticity of the healed tissue. Although collagen deposition restores most of the strength in the affected tissue, it is estimated that the new scar tissue is 20% weaker and less elastic than pre-injured tissue [22].

### 3.2. Fibrosis

Fibrosis is the excessive accumulation of connective tissue and ECM components [23]. In the healthy remodeling phase of wound healing, fibrosis is minimal. However, pathological fibrosis can result from an overly aggressive and unchecked healing response secondary to significant tissue injury, poor wound control, predisposed demographics, or an existing immunocompromised patient. In pathologic conditions, such as excessive conjunctival fibrosis, the normally efficient and orderly remodeling phase of wound healing is lost, and the conjunctival epithelium undergoes a state of chronic inflammation characterized by uncontrolled growth factor signaling (Figure 4).

TGF-β, released from macrophages, is the cardinal growth factor involved in the progression of fibrosis during wound healing [24]. Upon TGF-β stimulation, fibroblasts are activated and undergo transition into myofibroblasts, the key effector cells in fibrotic states [24]. Myofibroblasts in the conjunctiva are called Tenon fibroblasts; they augment fibrosis by depositing connective tissue, producing cross linking enzymes, and releasing MMPs during the proliferative stage of wound healing [25,26]. In a normal physiologic state, this process ends with the apoptosis of the myofibroblasts and the cessation of inflammation. However, acceleration into excessive fibrosis is mediated by exaggerated levels of various growth factors and cytokines, including TGF-β, interleukins, such as IL-1, IL-6, IL-10, and PDGF, as illustrated in the graphical abstract and in Figure 3 [27]. These factors ultimately lead to uncontrolled myofibroblast activation and, thus, pathologically excessive deposition of ECM [24].

In regard to glaucoma surgery, uncontrolled postoperative fibrosis is the main cause of procedure failure, resulting in excessive scarring, visual impairment, and subsequent progression of glaucoma. Table 2 (adapted from Yamanka et al., 2015) includes antifibrotic cytokines, growth factors, and signaling pathways relevant to preventing ocular fibrosis [28].

### 3.3. Wound Healing in Trabeculectomy

A trabeculectomy is a common filtering procedure performed in glaucoma patients [49]. A conjunctival incision is made followed by a partial-thickness scleral flap to expose the trabecular meshwork (Figure 5A). The AC is inserted, and a block of trabecular meshwork and SC are excised. After performing a localized iridectomy (Figure 5B), the scleral flap is reattached using interrupted sutures. Sponge-soaked MMC is applied under the conjunctiva for variable time followed by irrigation with a balanced salt solution. Figure 5C shows an ultrasound biomicroscopy (UBM) of a patient after trabeculectomy.

The AH flows under the scleral flap into the subconjunctival space, forming an aqueous humor reservoir commonly known as a filtering bleb. The formation of a shallow filtering bleb versus a large cystic bleb (which may restrict the flow of AH) is preferable for an optimal reduction in IOP. Unlike other ocular surgeries (cataract, retinal), where complete healing and restoration of the incised tissue is desired, the success of trabeculectomy depends on the optimal flow of AH under the scleral flap [28].

A trabeculectomy bleb undergoes four phases of postoperative wound healing. The first phase consists of an immediate inflammatory response and involves the recruitment of inflammatory cells, such as cytokines and growth factors. These inflammatory cells lay the foundation for the second phase, which involves the formation of highly vascularized granulation tissue, proliferation, and tissue repair. This phase may last through the second or third month after the surgery.

The third phase involves the activation, migration, and proliferation of episcleral fibroblasts, angiogenesis, and formation of collagen bundles. The final remodeling phase is characterized by the contraction of collagen bundles and scar tissue formation. The latter may impede flow of AH and its final absorption in the subconjunctival space. While healing under the scleral flap is important, the fibroblasts in the Tenon’s capsule are the main effector cells in the initiation and mediation of trabeculectomy wound healing and fibrotic scar formation [50].

The failure of glaucoma filtration surgery is mainly due to excessive subconjunctival wound fibrosis. Therefore, suppression of wound fibrosis is critical to maintain the smooth flow of AH [50]. Though the use of antifibrotic agents, such as MMC, have increased the success rate, there are still a number of complications, such as cystic blebs, dysaesthesia, wound leaks, blebitis, and endophthalmitis, which present challenges (Figure 6).

### 3.4. Wound Healing after Glaucoma Drainage Devices (GDDs) and Bleb-Forming MIGS

In certain high risk patients, such as those who had previously undergone a trabeculectomy, secondary glaucoma, or have African American heritage, the use of glaucoma drainage devices (GDDs) is preferred. The following section will focus on the wound healing process after commonly used GDDs and bleb-forming MIGS. Understanding the complex wound healing process following a GDD or bleb-forming MIGS procedure is crucial, since the healing success depends largely on how the eye responds after surgery [51]. In a GDD procedure, an implant is selected, which shares common features consisting of a biocompatible silicone tube and a plate of varying size that is positioned in the subconjunctival space [51]. Likewise, bleb-forming MIGS channel AH from the AC into the subconjunctival space (Figure 7A,B).

The tissue trauma caused by the aforementioned GDDs and bleb-forming MIGS (peritomy, cauterization, and suturing of the patch grafts) leads to the release of plasma proteins and other inflammatory cells, such as neutrophils, macrophages, and fibroblasts [52]. Additionally, AH has inflammatory properties and is known to contain growth factors (VEGF, FGF, PDGF-β, TGF-β) that can lead to a brisk fibrotic response in the subconjunctival space [53]. A 2013 study verified the presence of TGF-β2 in glaucomatous AH and also identified notably higher levels of chemokine (C-C motif) ligand 2 (CCL2; MCP-1) [54]. Controlling the inflammation caused by these factors is crucial to the success of GDDs and bleb-forming MIGS, since inflammation surrounding the endplate or AH outflow is the leading cause of implant failure [55]. The first-line treatment for decreasing postoperative inflammation is the use of corticosteroids, both topical and oral. Corticosteroids achieve their anti-inflammatory effects primarily by interfering with pro-angiogenic signal transduction pathways [56]. For example, a commonly used synthetic corticosteroid, dexamethasone, is an extremely strong anti-inflammatory agent, with effects up to six times more potent than prednisolone or triamcinolone and twenty-five times more than hydrocortisone [57]. These corticosteroids and broad-spectrum antibiotics are commonly administered subconjunctivally at the conclusion of the procedures [58]. Furthermore, topical application of corticosteroids is continued for 2–3 months following surgery to maintain a decreased inflammatory response.

As AH flows into the subconjunctival space, an excessive fibrotic reaction in the filtering bleb may result in bleb failure. The resultant encapsulation of the bleb impedes the AH outflow, resulting in elevated IOP [52]. The use of antimetabolites, namely MMC and 5-FU, have been efficacious in decreasing fibroblast proliferation following trabeculectomy, but their use in GDDs and bleb-forming MIGS is not widely accepted [59]. Some studies highlight the usage of MMC in the success of bleb-forming MIGS, but the benefits of MMC to GDD procedures remains unproven [60,61]. A 1995 study by Perkins et al. showed that while use of MMC with a double-plate Molteno implant showed a one-year success rate of 85% versus 20% in the control eyes, the two-year success rates were comparable for both groups [62,63]. A couple of years later, Lee et al. and Cantor et al. both concluded that adjunct use of MMC with Molteno implants did not offer significantly different outcomes from control groups at one-year post-surgery [64,65]. These studies showed a significantly higher incidence of complications in the MMC groups, including flat ACs and choroidal effusions. Additionally, a 2009 study demonstrating the adjunct use of MMC with the Ahmed glaucoma valve in infants with mostly primary congenital glaucoma (54.8%) or aphakic glaucoma (16.1%) showed that the MMC group had a significantly shorter bleb survival versus the control [66]. Currently, in Xen Gel Stent^®^ or PreserFlo^®^ MicroShunt procedures, surgeons either inject MMC or use MMC-soaked sponges [67]. However, it is still not commonplace to administer MMC during a GDD procedure.

MMC is potentially cytotoxic and may be associated with avascular and cystic blebs that are prone to complications, such as hypotony, blebitis, and endophthalmitis [68]. For this reason, there is a lot of interest in exploring the usage of different antimetabolites during MIGS. For example, in animal studies, valproic acid (VPA) has been used as an adjunct antifibrotic agent during implantation of the PreserFlo^®^ MicroShunt [69]. This study demonstrated that postoperative subconjunctival injections of VPA yielded significantly better outcomes than the control group treated with phosphate buffered saline. After two weeks post-surgery, the control group blebs failed, whereas the VPA group maintained diffused, fluid-filled blebs visible up to 28 days. Histology showed that in the VPA-treated groups, the subconjunctival stromal matrix was made of loosely arranged and thin criss-crossed ECM fibers, compared to the thick, disorganized fibers in the control group. This suggests that VPA improves bleb functionality by facilitating a less dense connective tissue structure. Additionally, VPA was found to suppress collagen and fibronectin gene expression, while enhancing the expression of factors disrupting TGF-β pathways. Another study comparing the concomitant usage of VPA and MMC with varying doses of MMC in a rabbit model of the PreserFlo^®^ MicroShunt found that the combination therapy was less cytotoxic when compared to MMC alone [70]. Moreover, the combination decreased VEGF and collagen gene expression more than MMC alone was able to. Together, these findings suggest that the usage of VPA as an antimetabolite in MIGS may reduce toxicity while more effectively managing the fibrotic response following implantation.

Although the use of steroids and antimetabolites is an integral aspect of managing inflammation and fibrosis in GDD and bleb-forming MIGS procedures, the biocompatibility of materials used in implants also plays a role in modulating wound healing. Most modern glaucoma devices are constructed from polypropylene (PP) and silicones, but their hydrophobic nature can lead to protein buildup and fibrosis [71]. To combat these complications, other materials, like gelatin and SIBS, have been innovatively used in the creation of the Xen Gel Stent^®^ and PreserFlo^®^ MicroShunt, respectively. Gelatin is a protein derived from collagen, and it is crosslinked with glutaraldehyde (GTA) to create the hydrophilic tube used in the Xen Gel Stent^®^ [72]. This combination of materials resulted in a stable implant that showed no signs of hydrolytic degradation. Moreover, implantation of these materials does not cause significant inflammation or a foreign-body tissue reaction [73]. In fact, in an early-stage pilot study, a collagen stent placed into the subconjunctival space without connecting to the AC or allowing AH flow, showed no fibrosis around it after six months [72]. However, a 2010 investigation comparing gelatin hydrogels cross-linked with GTA to those with 1-ethyl-3-(3-dimethyl aminopropyl)carbodiimide (EDC) in rat iris pigment epithelium revealed that the EDC-treated groups exhibited lower levels of cytotoxicity, IL-1β, and TNF-α levels than GTA-treated ones. Furthermore, GTA groups demonstrated significant inflammation, suggesting EDC as a biocompatible alternative for GTA. However, further research is needed for its application in glaucoma implants. In addition, a 2006 study examining the usage of SIBS in a drainage implant instead of silicone demonstrated noncontinuous collagen deposition with no macrophages or myofibroblasts visible around the SIBS tube versus collagen deposition and myofibroblast differentiation induced by silicone [74]. A study conducted in 2022 involving fifteen New Zealand White rabbits that were implanted with PreserFlo^®^ MicroShunts revealed the presence of a wide variety of cells, including polymorphonuclear leukocytes, myofibroblasts, and foreign body giant cells within the bleb and around the microshunt postoperatively [75]. These findings suggest that although the implantation of the SIBS MicroShunt has been efficacious as a bleb-forming MIGS, the presence of certain fibrotic factors may affect long-term outcomes.

Despite the innovation of new postoperative treatments and biocompatible implant materials, fibrosis continues to be a limiting factor in many glaucoma surgeries. Thus, further studies are needed to continue research on novel antifibrotic drugs and materials.

### 3.5. Current Glaucoma Wound Healing Agents

A common surgical complication after glaucoma surgery is the formation of scarring, which impedes the flow of AH. Therefore, treatment modalities have focused on reducing fibroblast production in order to decrease fibrosis postoperatively [76]. In the early 1990s, MMC and 5-FU were tested, and both showed high effectiveness [77].

MMC is a natural alkaloid synthesized from *Streptomyces caespitosus*, a species of actinobacteria [78]. It reduces fibroblast collagen synthesis by inhibiting DNA-dependent RNA synthesis and inducing DNA crosslinking (Figure 8) [35]. The crosslinked DNA segments block key DNA metabolism steps, including the replication and transcription of fibroblasts, which reduces collagen deposition and ultimately decreases the extent of scar formation at the subconjunctival site [79]. As MMC is most efficiently converted to its active form in Tenon’s fibroblasts compared to fibroblasts from other parts of the body, it is widely used as an agent of choice during filtration surgery. In a 1992 study on human Tenon’s capsule tissue, MMC administration led to the inhibition of fibroblast proliferation by 31.3% [78]. Additionally, MMC is significantly more potent than 5-FU, and is currently the agent of choice [78].

5-FU is a pyrimidine analog that selectively inhibits both DNA and RNA synthesis, thus halting cellular proliferation and inducing direct cytotoxicity [80]. It is converted to three primary active metabolites: fluorodeoxyuridine monophosphate (FdUMP), fluorodeoxyuridine triphosphate (FdUTP), and fluorouridine triphosphate (FUTP), as shown in Figure 9. Its conversion to FdUMP forms a stable complex with an enzyme called thymidylate synthase, which inhibits DNA replication and repair [80]. In a 2008 study assessing 5-FU’s use as an antimetabolite during trabeculectomy, it was shown to significantly reduce the risk of surgical failure in patients undergoing initial trabeculectomy, with a success rate of 81.6% (compared to 20.4% in controls) after 6 months [80].

It is well established that the usage of 5-FU and MMC has significantly improved success rates in glaucoma surgery [77]. However, these agents can cause widespread cell death, which increases the risk of several complications, such as prolonged subconjunctival hemorrhage and the formation of thin-walled avascular blebs that are prone to leakage and infection [81]. Therefore, the search for less toxic antifibrotic agents is crucial in reducing postoperative complications.

Secondly, controlling inflammation after glaucoma surgery is also of utmost importance for bleb survival. Topical corticosteroid agents have been used to control inflammation in the postoperative period [82]. They are thought to stimulate a steroid receptor in the nucleus of each cell, resulting in the widespread modification of up to 6000 genes within a few hours of its exposure [83]. Their anti-inflammatory property is largely mediated by the suppression of leukocyte concentration and vascular permeability (characterized by the inflammatory phase of wound healing). Consequently, this leads to decreased local tissue damage, reduced release of pro-fibrotic mediators, and less production of fibrin clots (involved in the hemostasis stage of wound healing) [84]. Broadway et al. were the first to show a significant reversal in macrophages, lymphocytes, and mast cells of conjunctival tissues after one month of preoperative steroids; their surgical success rates were also improved from 50% to 81% [85].

In some patients, steroid response (elevated IOP) is a significant side effect after prolonged topical corticosteroid usage. Its prevalence is approximately 18% to 36%, but it has been reported to be as high as 92% in patients with POAG [86,87]. Thus, clinicians must be watchful for elevated IOP after corticosteroid use and manage it appropriately with anti-glaucoma medications.

Bevacizumab is a recombinant humanized anti-VEGF immunoglobulin, which was initially used in the treatment of metastatic cancers, but which is now widely used in ophthalmology for proliferative diabetic retinopathy, exudative macular degeneration, macular edema, retinal vein occlusions, and neovascular glaucoma [88]. VEGF encourages angiogenesis (proliferative stage of wound healing), which ultimately results in fibrosis [88]. In a study at the Catholic University of Korea, increased amounts of VEGF were found in the vitreous and AH in glaucoma patients undergoing trabeculectomy. This prompted the authors to try anti-VEGF agents to reverse postoperative scarring [89]. Later, in 2012, Ghanem published a study using 55 patients to compare the use of subconjunctival bevacizumab versus a placebo in patients undergoing a primary trabeculectomy with MMC [90,91]. At a one-year follow up, he found a statistically significant reduction in vascularity of the filtering bleb in the bevacizumab + MMC group compared to the placebo group [90]. Table 3 shows a summary of each agent’s mechanism of action and administration. 

### 3.6. Landmark 5-FU and MMC Studies


**Author/Year/Country**

**Results**
Kitazawa Y. et al., 1991. Japan[92]Thirty-two patients undergoing trabeculectomy were assigned to receive either MMC (seventeen eyes) or 5-FU (fifteen eyes). The mean preoperative IOPs (mmHg) were 28.7 ± 7.9 (MMC) and 32.7 ± 10.0 (5-FU). At the final post-op visit, the mean postoperative IOPs were 8.6 ± 3.8 (MMC) and 12.3 ± 4.2 (5-FU). The incidence of corneal complications was lower in the MMC group (12%) compared to the 5-FU group (53%).Katz GJ et al., 1995. USA[93]In a high-risk filtration study, 20 patients received MMC and 9 received 5-FU. The mean preoperative IOP’s (mmHg, MMC vs. 5-FU) were 32.6 ± 10.5 and 31.5 ± 9.8, respectively (*p* = 0.78). At 32 months, the postoperative IOP’s were, similarly, 9.0 ± 4.9 vs. 16.3 ± 4.8 (*p* = 0.0003). The MMC group required fewer medications for IOP control (0.5 vs. 1.6) (*p* = 0.01).Lamping et al., 1995. USA[94]A total of 74 pseudophakic patients with glaucoma underwent trabeculectomy, and received either 5-FU (40 eyes) or MMC (40 eyes). Preoperative IOP’s (mmHg, MMC vs. 5-FU) were 30.6 vs. 31.5, respectively. At 12 months post-op, the IOP’s were, similary, 12.8 vs. 14.8 mmHg (*p* = 0.001). The MMC-treated eyes required fewer IOP-lowering medications (0.6) compared to 5-FU-treated eyes (1.05) (*p* = 0.03).Zadok D et al., 1995. Israel[95]This trabeculectomy study compared postoperative subconjunctival injections of 5-FU (19 eyes) with single intraoperative application of subconjunctival MMC (20 eyes). At 6 months, IOPs averaged 10.9 mmHg (MMC-treated eyes) vs. 14.2 mmHg (5-FU-treated eyes) (*p* = 0.14). The MMC-treated group was on fewer medications (0.3 vs. 1.1, *p* < 0.001).Cohen et al., 1996. USA[96]In a combined cataract and trabeculectomy study, 72 eyes were randomized to MMC (0.5 mg/mL) vs. a placebo. At 6 months, significantly fewer medications were required for the MMC group (0.5 vs. 1.2; *p* = 0.002). Similarly, at 12 months, the MMC group had significantly reduced mean IOP (7.65 mmHg vs. 3.84 mmHg; *p* = 0.001). However, the MMC group showed large filtering blebs and more frequent wound leaks.Costa et al., 1996. Brazil[97]A total of 28 eyes with advanced POAG were given either MMC (0.2 mg/mL) or saline solution intraoperatively for 3 min. Mean IOPs were significantly lower in the MMC group compared to the controls at the final post-op visit (*p* = 0.001). The IOP (mmHg) was ≤15 in 85.7% (MMC) vs. 28.6% (control, *p* = 0.002). Choroidal effusions (35.7% vs. 14.3%, *p* = 0.0065) and shallow AC (35.7% vs. 7.1%) were more common in the MMC group.Carlson et al., 1997. USA[98]In a combined phacoemulsification and trabeculectomy procedure, 29 patients received either MMC [0.5 mg/mL] or a placebo. Pre-op IOPs (mmHg) were 18.4 ± 2.7 (MMC) vs. 19.1 ± 4.0 (placebo). At 8 months, MMC-treated eyes had a lower average IOP (12.3 ± 1.6) compared to the placebo-treated eyes (15.2 ± 1.5). At 12 months, IOPs averaged 12.6 ± 1.0 (MMC) and 16.2 ± 1.5 (placebo). On average, the MMC group had lower post-op IOP levels than the placebo group (*p* = 0.04).Singh et al., 1997. USA[99]A total of 101 eyes of black Ghanian patients with POAG were treated with either 5-FU and MMC after trabeculectomy. The 5-FU group (50.0 mg/mL for 5 min) had 57 patients, and the MMC group (0.5 mg/mL for 3.5 min) had 44 patients. Overall mean pre-op IOP (mmHg) was 30.1. Patients receiving MMC (IOP = 14.7) had a lower mean postoperative IOP than those receiving 5-FU (IOP = 16.7; *p* = 0.05).Singh et al., 1997. USA[100]In a black West African population, 81 eyes were divided to receive MMC or 5-FU during trabeculectomy. A total of 37 received 5-FU (50 mg/mL for 5 min) and 44 received MMC (0.4 mg/mL for 2 min). Pre-op IOP (mmHg) was 30.7 (MMC) vs. 32 (5-FU). The mean post-op IOP was 13.7 (MMC) vs. 16.3 (5-FU, *p* = 0.05).Andreanos et al., 1997. Greece[101]The study assessed MMC in 46 patients (26 M + 20 F) undergoing a repeat trabeculectomy. Patients were randomly assigned to MMC (24) vs. control group (22). Pre-op IOPs (mmHg) ranged from 27 to 38. Post-op complications were higher in the MMC group, including choroidal effusion (8.3% vs. 0%) and shallow AC (29.2% vs. 13.6%). Mean IOP (≤20 mmHg after 18 months) was 83.3% in the MMC group compared to 63.6% in the control group.Singh et al., 2000. USA[102]In this trabeculectomy study, 54 eyes received MMC (0.4 mg/mL for 2 min) and 54 eyes received 5-FU (50 mg/mL for 5 min). At 3 years post-op, there was no statistically significant difference between the two groups for mean preoperative IOP, or post-op interventions/complications.DeBry et al., 2002. USA[68]In this trabeculectomy study involving 239 eyes, a Kaplan–Meier analysis suggested 5-year probabilities of developing endophthalmitis (7.5%), bleb leaks (17.9%), and blebitis (6.3%). Trabeculectomy with MMC was associated with significant morbidity, and the risk of complications reached 23% at 5 years.WuDunn et al., 2002. USA[103]A total of 115 eyes underwent trabeculectomy [57 eyes (5-FU) and 58 eyes (MMC)]. The mean preoperative IOP (mmHg) was 24.3 (5-FU) vs. 21.9 (MMC), with no statistical significance (*p* = 0.09). At 12 months, 94% of 5-FU eyes and 89% of MMC eyes reached the target IOP of 21 mmHg (*p* = 0.49).Sisto et al., 2007. Italy[104]A total of 40 eyes with neovascular glaucoma were divided to receive post-op 5-FU (18) vs. intraoperative MMC (22) after filtration surgery. Pre-op IOPs (mmHg) were 40.4 ± 10.3 (5-FU) and 42 ± 11.3 (MMC), respectively. The mean follow-up period was 35.8 (5-FU) and 18.6 (MMC) months. Although the mean IOP significantly decreased in both groups [from 40 to 14.7 (5-FU) group (*p* < 0.0001); vs. 42 to 29.9 (MMC) group (*p* = 0.0006)], the difference between the two groups was not significant.Mostafaei et al., 2011. Iran[105]A total of 40 patients with high-risk open angle glaucoma received either MMC or 5-FU. Mean preoperative IOPs (mmHg) were 30.6 (5-FU) and 31.2 (MMC), respectively. At 6 months, the mean IOPs postoperatively for 5-FU (13.6) and MMC (11.4) were similar. The relative success of 5-FU vs. MMC was 0.93 [95% CI: 0.8–1.1].Fendi et al., 2013. Brazil[106]A meta-analysis of 5 randomized controlled clinical trials comprising 416 patients comparing MMC against 5-FU was carried out. Pre-op IOP was ≥21 mmHg in both groups. Lower IOPs (mean difference 2.17 mmHg) and higher success rates were observed in the MMC arm (92%) than in the 5-FU arm (84.2%, *p* = 0.01).

### 3.7. Experimental Wound Healing Agents

#### 3.7.1. Nanoparticles

Nanomedicine encompasses the comprehensive regulation, repair, and improvement of human biology at the molecular level [107]. This is achieved by engineered nanodevices and nanostructures that operate in parallel at the single-cell level, with the goal of achieving desired medical benefits [108].

This new technology has prompted the need to develop newer drug delivery systems that allow for the gradual and sustained release of a drug, combined with improving bioavailability and minimizing complications (Figure 10). Many new nanoparticles composed of different structures (hollow, solid, or porous), shapes, and sizes have been developed. They contain or encapsulate certain molecules, such as drugs, DNA, RNA, or antibodies [109].

Common nanodelivery systems include nanoparticles, nanodiamonds (NDs), dendrimers, liposomes, and other devices. Drugs are incorporated into these nanomaterials through encapsulation or surface conjugation. Encapsulated drugs are released as the nanomaterials disassemble at the intended site, while conjugated drugs are released when the bond between the nanomaterial and drug is cleaved at the target site [110]. These nanomaterial-based drug delivery strategies have the potential to overcome limitations of conventional glaucoma treatments. Furthermore, incorporating inorganic nanoparticles into a hydrogel may enhance efficacy at the same or less dosage [109].

#### 3.7.2. Targeting mRNAs

Noncoding RNAs, including long noncoding RNAs (lncRNAs; LINC) and microRNAs (miR; miRNA), are increasingly being studied as key regulators of scarring in bleb formation after glaucoma filtering surgery. Both miRNA-200a and miRNA-200b are believed to promote fibrosis in the glaucoma filtering tract. Studies have shown that the expression of miR-26a in fibrotic bleb tissue varies and is downregulated compared to controls [111]. Enhanced expression of miR-200b has been observed in trabecular meshwork cells treated with TGF-β during post-trabeculectomy scarring [112]. Further investigations by Drewry et al. have shown that miRNA-200b affects the activity of two pathways that regulate cell proliferation, namely p27/kip1 and RND3. They have also shown that inhibition of phosphatase and tensin homolog (PTEN) gene, an inhibitor of the PI3K/Akt pathway (cell growth, proliferation, and migration), results in increased expression of the profibrotic proteins P13K, Akt, α-SMA, and fibronectin [113]. However, the specific genes influenced by miR-200b and their downstream effects remain unclear [113]. Overall, a more in-depth exploration of noncoding RNAs is necessary to comprehend their roles in the development of glaucoma and the identification of potential therapeutic targets [114].

Yu et al. reviewed potential anti/pro-fibrotic noncoding RNA agents that may be used in glaucoma filtering surgery (Table 4) [114].

#### 3.7.3. Infliximab

Infliximab is a chimeric monoclonal antibody that targets tumor necrosis factor (TNF)-α and which is composed of both mouse and human elements (human–murine IgG1). TNF-α acts as a local regulator for leukocytes and endothelial cells, functioning through paracrine and autocrine pathways and influencing immunological and inflammatory cascades [124]. Infliximab works by binding to TNF-α, thereby blocking NF-kB (transcription factor for the inflammatory process) migration, resulting in a decrease in the production of pro-inflammatory cytokines, such as IL-1 and IL-6, and adhesion molecules [125,126]. Therefore, infliximab may be a potential agent in modulating surgical fibrosis.

### 3.8. Future Directions

To improve the reliability and validity of the findings presented in this review, additional comparative research involving promising new antimetabolite agents is warranted. These future agents include anti-TGFβ agents (lerdelimumab, fresolimnumab, pirfenidone), kinase inhibitors (Nintedanib), anti-TNF-α agents (infliximab), beta-radiation, and nanotechnology-based drug delivery systems. In a study by Shao et. al, researchers concluded that beta radiation during trabeculectomy can reduce fibroblast proliferation and increase the success of glaucoma filtration surgery, but it may also lead to cataract formation [76]. Similarly, nanotechnology-based drug delivery systems have shown great promise in post-surgical wound healing [76]. Sustained-release implants, hydrogels, liposomal systems, and nanoparticles have been explored for targeted delivery and enhanced drug residence time, preventing rapid clearance and improving efficacy of antifibrotic agents [76]. While these new agents show great potential, further studies need to be conducted to optimize the delivery methods and to reduce complications.

## 4. Conclusions

It is well known that the long-term efficacy of glaucoma surgery is reduced by fibrosis, scar formation, and uncontrolled wound healing. Conventional adjuncts used for mitigating post-surgical fibrosis, such as corticosteroids and anti-fibrotic agents, have unpredictable outcomes and side effects. The ongoing research using promising experimental wound healing agents and new drug targets to prevent fibrosis may improve glaucoma surgery outcomes.

## Figures and Tables

**Figure 1 bioengineering-11-00446-f001:**
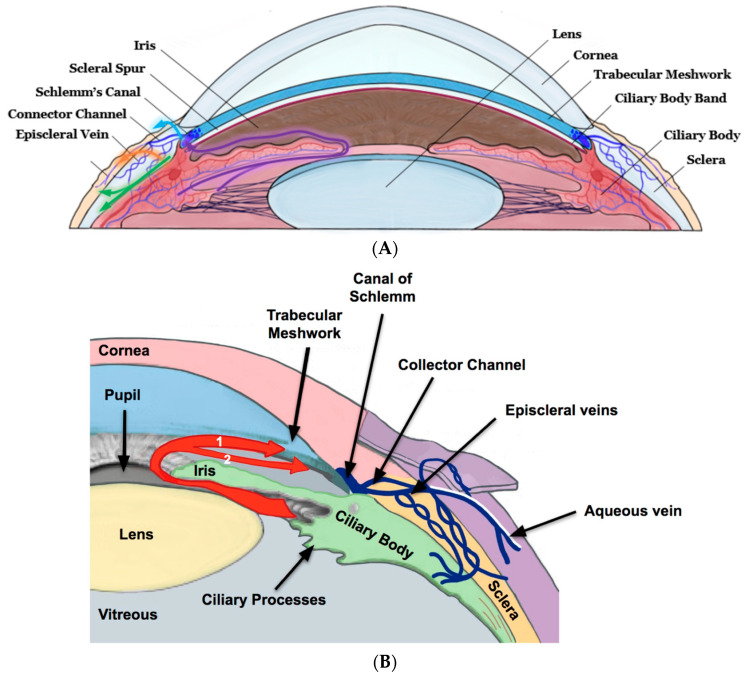
(**A**) Cross-section of an eye illustrating the AH flow dynamics. AH is formed by the ciliary body and flows through the pupil into the anterior chamber (AC). The drainage of AH is mainly via the conventional [TM, SC, and EV] pathway and the non-conventional [US-UV] pathway. (**B**) Higher magnification of (**A**). Red arrow #1 denotes AH flow from the trabecular meshwork through the Schlemm’s canal, collector channels, aqueous veins, and into the episcleral veins for drainage into the bloodstream. The uveoscleral pathway (red arrow #2) shows AH flowing directly through the ciliary muscle to the suprachoroidal space, and out through the sclera, eventually reaching general circulation.

**Figure 3 bioengineering-11-00446-f003:**
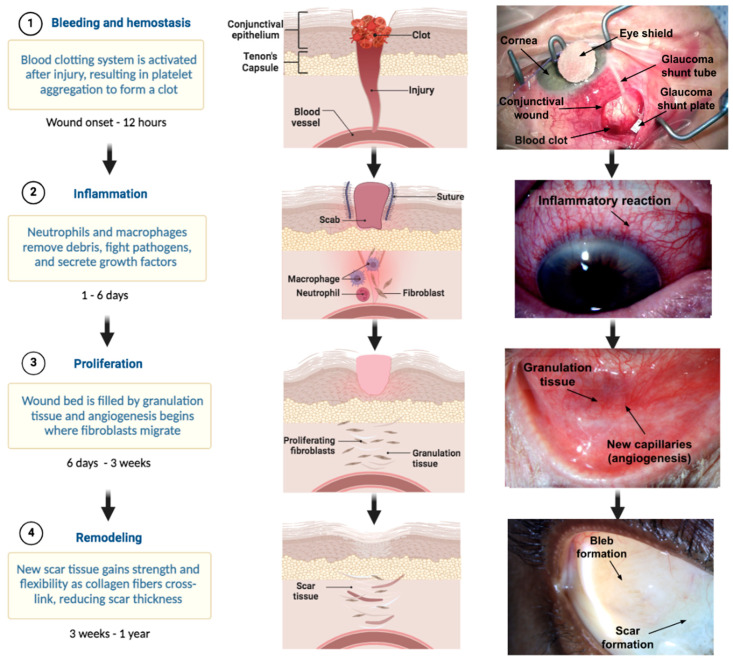
An overview of the chronology seen in the general healthy wound healing process in the eye: From left to right, this figure shows (1) hemostasis, (2) inflammation, (3) proliferation, and (4) remodeling.

**Figure 4 bioengineering-11-00446-f004:**
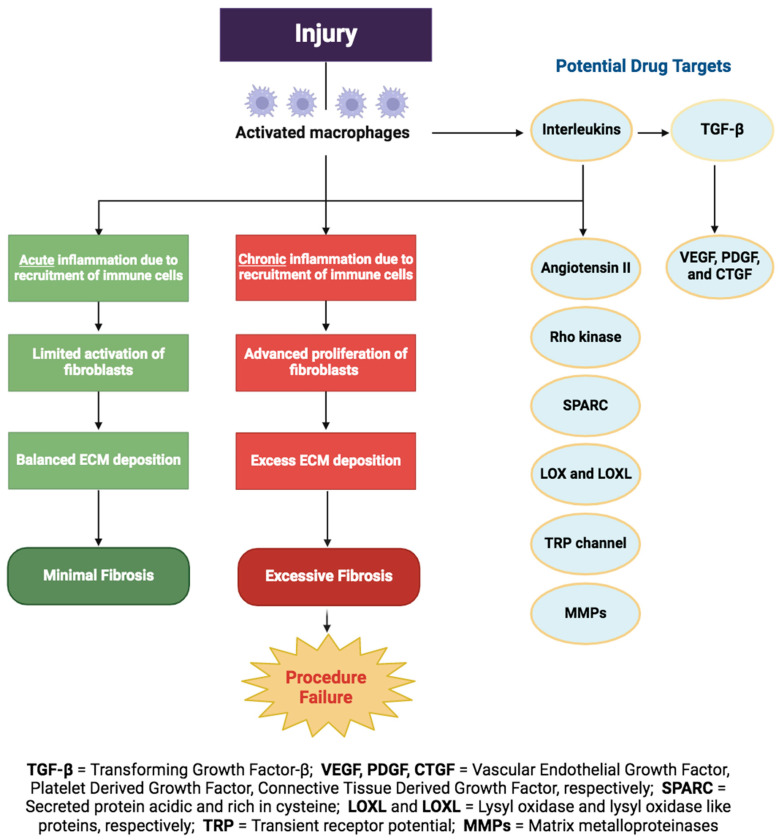
Flow diagram showing two distinct healing outcomes of fibrosis: minimal or excessive fibrosis after glaucoma surgery, as well as potential drug targets.

**Figure 5 bioengineering-11-00446-f005:**
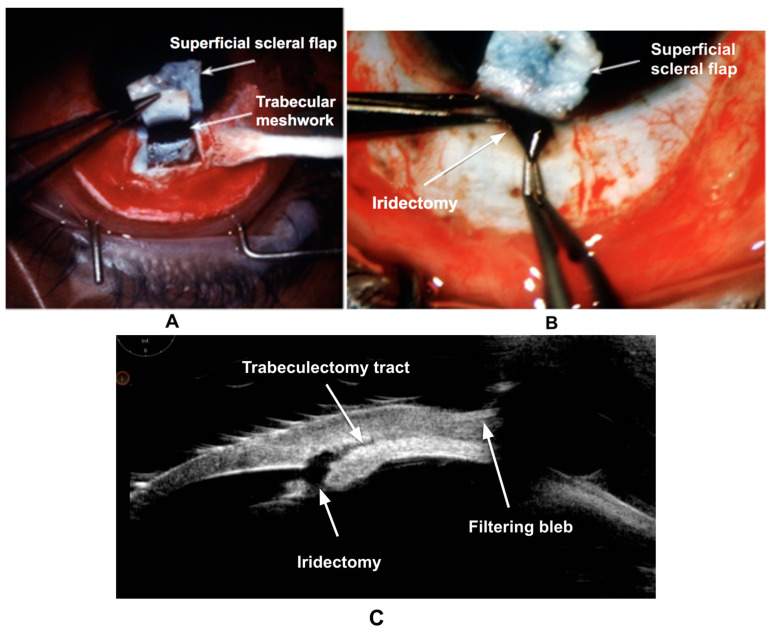
(**A**–**C**). Steps of trabeculectomy: conjunctival incision, superficial scleral flap, removal of trabecular meshwork/SC block, and iris (iridectomy). (**C**) shows an ultrasound biomicroscopy (UBM) after trabeculectomy.

**Figure 6 bioengineering-11-00446-f006:**
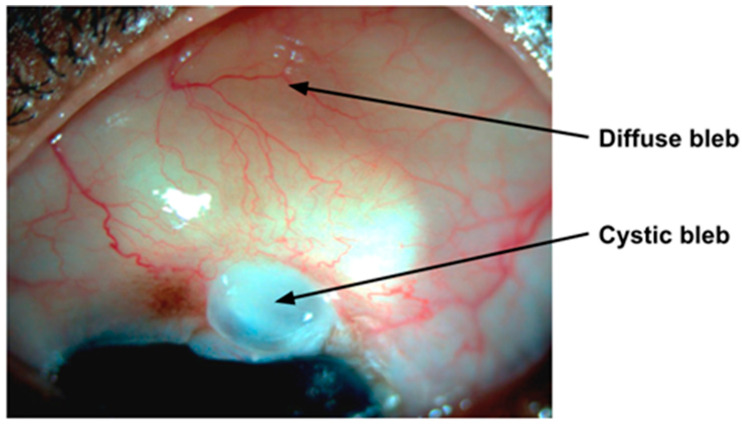
A cystic bleb at the limbus and a diffuse bleb formed after trabeculectomy.

**Figure 7 bioengineering-11-00446-f007:**
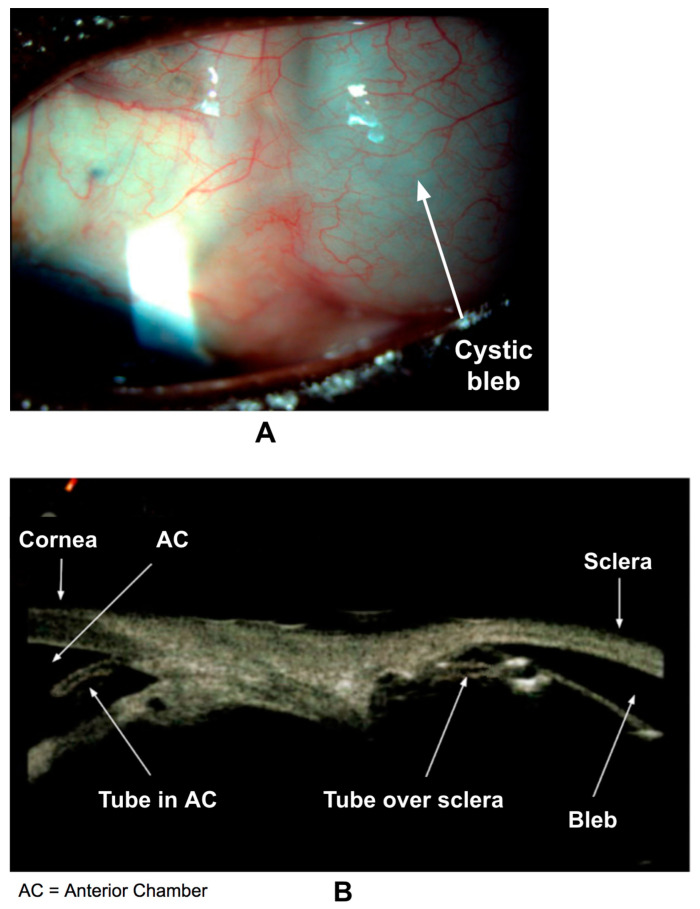
(**A**) A large, encysted bleb superolaterally in the left eye, formed after the insertion of a GDD. (**B**) Ultrasound biomicroscopy of the anterior segment shows the tip of the GDD in the anterior chamber. Posteriorly, the GDD tube is seen laying on the sclera, and a large filtering bleb is clearly visible.

**Figure 8 bioengineering-11-00446-f008:**
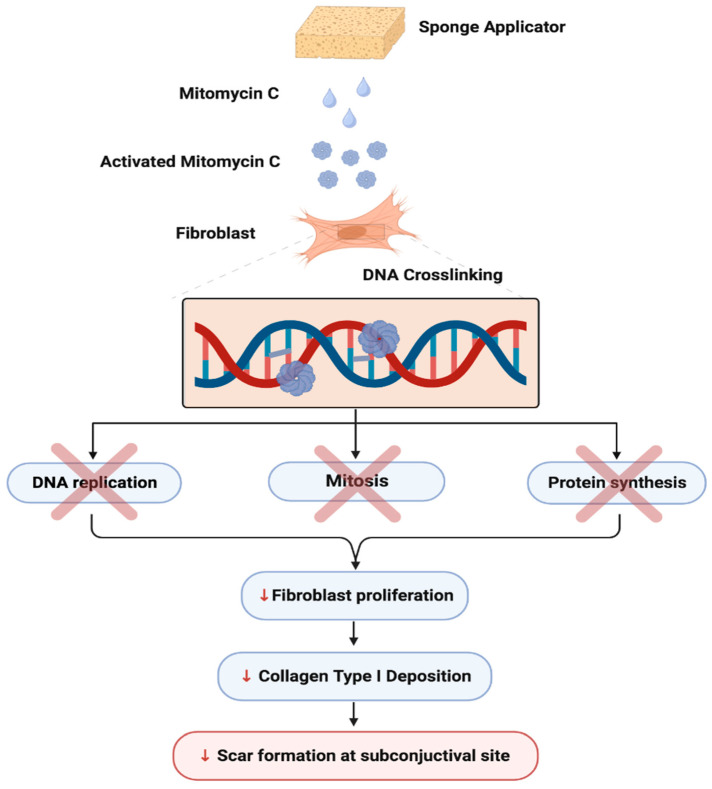
The mechanism of action and effects of mitomycin C.

**Figure 9 bioengineering-11-00446-f009:**
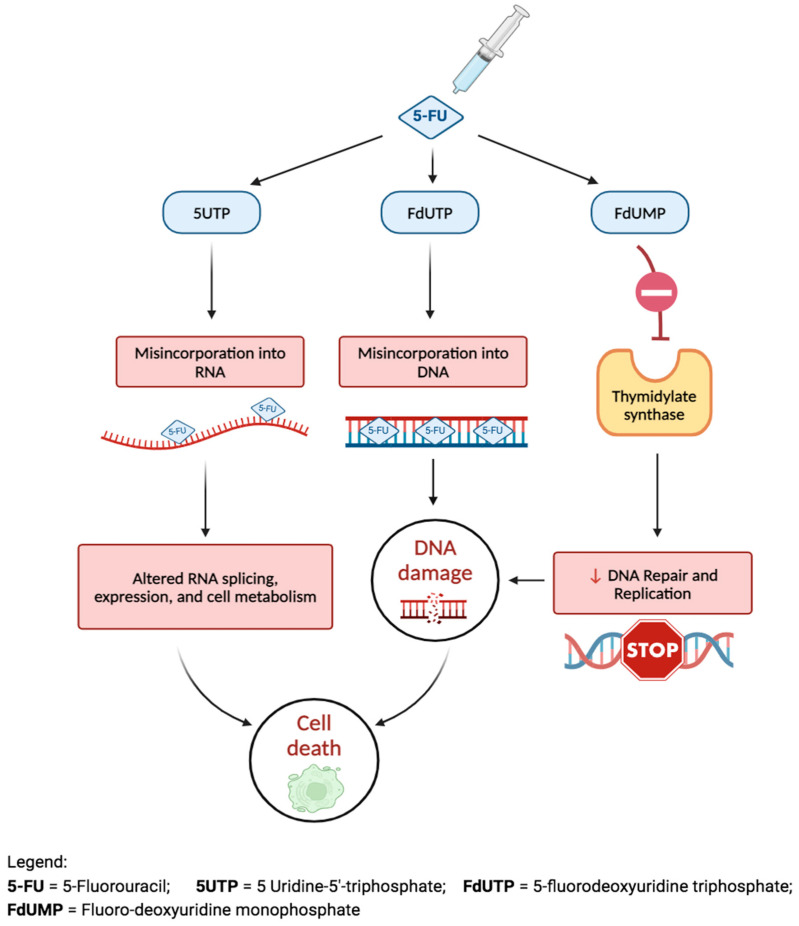
The mechanism of action and effects of 5-fluorouracil.

**Figure 10 bioengineering-11-00446-f010:**
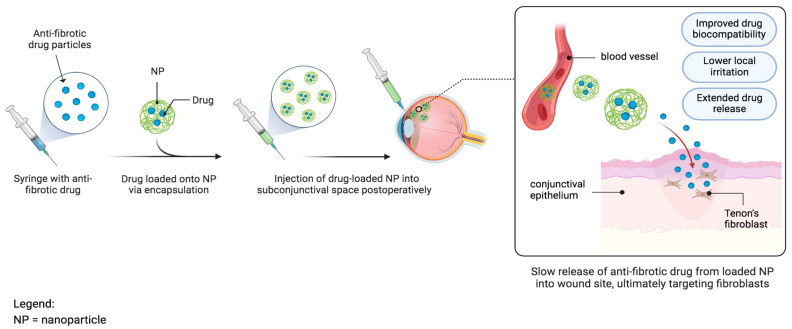
A schematic demonstrating nanotechnology-mediated drug delivery involving an antifibrotic drug encapsulated in a nanoparticle.

**Table 1 bioengineering-11-00446-t001:** PICOS Criteria for Inclusion of Studies.

Parameter	Description
Population	Patients with glaucoma regardless of the site.
Intervention	Incisional/filtration glaucoma procedures, with or without antifibrotic agents.
Comparison	Results of patients who underwent glaucoma surgery with and without antifibrotic agents.
Outcomes	Quality of IOP control, postoperative complications, visual acuity.
Study Design	Randomized or nonrandomized controlled (or uncontrolled).

**Table 2 bioengineering-11-00446-t002:** Antifibrotic targets and their mechanism of action, adapted from Yamanka et al. [28].

Antifibrotic Targets	Mechanism of Action	Applications
IL-1 [29]	IL-1 controls integrin expression in leukocytes and endothelial cells.	1-methyl hydrazino analogs are an excellent IL-1 blocker and reduce inflammation.
IL-6 [28,30]	IL-6 stimulates B-cell differentiation, T-cell activation, and immunoglobulin production.	Tocilizumab is an anti- IL-6 receptor antibody, which, in a rheumatoid arthritis clinical study, reduced inflammation and fibrosis.
IL-7 [31,32]	IL-7 is a profibrotic growth factor and activates signaling that suppresses fibroblast-driven ECM expression.	In a septic shock trial, IL-7 application restored CD4+ and CD8 cell count.
IL-10 [33,34,35]	IL-10 is an anti-inflammatory cytokine which reduces production of inflammatory cytokine mRNA.	In a mice study, IL-10 increased the number of neutrophils and monocytes.
IL-22 [36,37,38]	IL-22, a pro-inflammatory cytokine, upregulates acute phase proteins.	In a hepatitis clinical trial, IL-22 protected against epithelial cell injury and reduced inflammation.
Anti-VEGF [39]	VEGF is a potent mediator of angiogenesis, vasculogenesis and vascular endothelial cell permeability.	Anti-VEGF therapies inhibit vascular endothelial growth factor, thus preventing angiogenesis and the disruption of the blood–retinal barrier.
Platelet-derived growth factor (PDGF) [40]	The PDGF family consists of disulphide-linked dimers and induces proliferation of macrophages and fibroblasts migration into a wound site.	ARC126 and ARC127 are PDGFβ inhibitors, and they reduced both epiretinal membrane formation and retinal detachment.
Connective tissue growth factor (CTGF) [41]	CTGF is a fibrogenic cytokine upregulated by TGF-β causes persistent fibrosis through CTGF.	Targeting either CTGF or TGF-β signaling may reduce scar tissue formation.
Matrix metalloproteinases (MMPs) [28,42]	MMPs are a group of proteolytic enzymes which degrade most extracellular matrix proteins during wound remodeling.	Administration of GM6001, an MMP inhibitor, reduced scar formation after glaucoma surgery in rabbits.
Lysyl oxidase (LOX) and lysyl oxidase-like proteins (LOXL) [28,43]	Lysyl oxidase (LOX) and lysyl oxidase-like (LOXL) are ECM enzymes which crosslink collagen and elastin, leading to fibrosis.	Anti LOXL2 monoclonal antibody (GS-607601) reduced inflammation and fibrosis after glaucoma surgery in rabbits.
Rho kinase inhibitors [28,44]	ROCK 1 and 2 are downstream components of Rho-GTPase Rho mediated signaling and play an important role in cytoskeletal organization controlling cellular morphology migration and motility. Rac1 is a low-molecular-weight Rho GTPase.	In a lab experiment, inhibiting Rac1 with NSC23766 or siRNA achieved reduction in conjunctival tissue fibrosis and collagen matrix contraction.
Secreted protein acidic and rich in cysteine (SPARC) inhibitors [28,45]	SPARC is a 43 kDa collagen-binding matricellular glycoprotein that modulates cellular interactions with the surrounding ECM. SPARC contributes to ECM organization and cell migration.	In an in vitro experiment, SPARC knockdown resulted in TGFβ2-driven upregulation of Type I collagen, and fibronectin expression was suppressed. Reducing SPARC expression may suppress subconjunctival fibrosis.
Angiotensin II [28,46]	Angiotensin II is an effector molecule and causes ocular fibrosis. Activation of NF-κB by angiotensin II leads to the survival of corneal myofibroblasts, and, consequently. fibrosis.	In lab experiments, angiotensin-converting enzyme inhibitors (ACE II s) and angiotensin receptor (AT2) antagonists effectively suppressed vascular damage.
Transient receptor potential (TRP) channel antagonists [28,47]	The TRP channels are activated by multiple endogenous and external stimuli and mediate several wound healing functions. Their receptor-induced responses include cell proliferation and migration, along with immune cell activation, tissue infiltration, and fibrosis.	In an alkali-burn mouse wound healing model, treatment with a TRPV1 antagonist effectively suppressed fibrosis. Additionally, in vitro experiments using ocular fibroblasts demonstrated that the TRPV1 antagonist inhibited the transdifferentiation of myofibroblasts.
Transforming growth factor-β (TGF-β) inhibitor [28,48]	TGF-β plays a significant role as an effective mediator in the development of scar tissue in the eye.	In lab experiments, tranilast suppressed TGF-β activation and resulted in the suppression of collagen production. In vitro experiments using siRNA to suppress the TGF-β type II receptor gene demonstrated both suppression of fibronectin production and inhibition of cell migration.

**Table 3 bioengineering-11-00446-t003:** Current wound healing agents.

Agent	Mechanism of Action	Administration
Mitomycin C (MMC) [79]	An alkaloid, produced by *Streptomyces caespitosus*; works by inhibiting DNA-dependent RNA synthesis and triggering apoptosis.	Either via MMC-soaked sponge or subconjunctival injection postoperatively.
5-fluorouracil (5-FU) [80]	A pyrimidine analog, interferes with ribosomal RNA synthesis; diminishes episcleral scar formation by inducing apoptosis of fibroblasts in Tenon’s capsule.	Similar to MMC.
Corticosteroids [84]	Reduce the expression of cytokines, such as TNF-alpha, IL-1, IL-2, IL-10, and IL-12, which decrease the number of tissue macrophages and blood monocytes during the inflammatory phase of wound healing.	Topical, subconjunctival injection, or oral perioperatively.
Bevacizumab [89]	Selectively binds to and blocks circulating VEGF to reduce micro-angiogenesis, thereby limiting the blood supply to scarred granulation tissue during the proliferative phase of wound healing.	Subconjunctival injection postoperatively.

**Table 4 bioengineering-11-00446-t004:** A compiled summary of studies associated with noncoding RNAs, adapted from Yu et al., 2022 [114].

Noncoding RNAs	Authors, Year, Country	Summary	Pro/Anti-Fibrotic Role
miR-26a	Wang et al., 2018, China [115]	miR-26a is significantly downregulated in filtering tract scars and is inversely correlated with connective tissue growth factor (CTGF) mRNA levels.	Anti-fibrotic
miR-29b	Ran et al., 2015, China [116]	TGF-β2 stimulates the proliferation of human tenon fibroblasts (HTF) by suppressing miR-29b expression, which is regulated by Nrf2.	Anti-fibrotic
miR-139	Deng et al., 2019, China [117]	Overexpression of miR-139 effectively counteracted the TGFβ1-induced increase in collagen I and α-smooth muscle actin levels, as well as HTF proliferation.	Anti-fibrotic
miR-200a	Peng et al., 2019, China [118]	miR-200a is reduced, while FGF7 is increased in glaucoma. miR-200a has a protective function on the glaucomatous optic nerve injury through its effect by suppressing the MAPK signaling pathway mediated by FGF7.	Anti-fibrotic
miR-200b	Tong et al., 2019, China [119]	The induction of fibrosis in HTFs occurs through TGF-β1-mediated miR-200b by suppressing the PTEN gene signaling pathway.	Pro-fibrotic
miR216b	Xu et al., 2014, China [120]	miR-216b directly targeted and decreased the expression of Beclin 1, a pro-apoptotic molecule. In HTFs treated with hydroxycamptothecin, miR-216b regulates both autophagy and apoptosis by modulating Beclin 1.	Pro-fibrotic
Lnc H19	Zhu et al., 2020, China [121]	TGF-β induced the expression of H19 in HTFs, and suppressing H19 inhibited the effects of TGF-β. The findings suggest that H19 modulates β-catenin expression via miR-200a in TGF-β-treated HTFs. Therefore, suppressing H19 may result in attenuation of scar after glaucoma surgery.	Pro-fibrotic
Lnc NR003923	Zhao et al., 2019, China [122]	Inhibiting NR003923 expression in HTFs resulted in the suppression of cell migration, proliferation, fibrosis, and autophagy induced by TGF-β.	Pro-fibrotic
LINC00028	Sui et al., 2020, China [123]	In HTFs treated with TGFβ1, the decrease in LINC00028 expression inhibits migration, proliferation, invasion, epithelial-mesenchymal transition, fibrosis, and autophagy.	Pro-fibrotic

## Data Availability

No new data were created or analyzed in this study. Data sharing is not applicable to this article.

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
