# Peer review of "Wound Modulations in Glaucoma Surgery: A Systematic Review"

_bioengineering, 2024, doi:10.3390/bioengineering11050446_

Round 1
Reviewer 1 Report
Comments and Suggestions for Authors
I found this manuscripg very intersting manuscript and this systematic review was siutabla for publication .
Author Response
Thank you so much for your comments. We appreciate your kind remarks and your approval of our manuscript.
Reviewer 2 Report
Comments and Suggestions for Authors
see attached

The English is OK.
Author Response
We really appreciate your comments! Please see the attachment.

Reviewer 3 Report
Comments and Suggestions for Authors
This is an excellent review describing techniques and pitfalls of glaucoma surgical treatments. Having myself had a past interest in this field, I may only suggest a further addition to their list of potential innovative treatments to control filtration surgery. The work published in 2023 by Minnelli et al. (DOI: 10.3390/ph16040594) described the effects of Meldonium on the control of IOP and simultaneously interfering with the fibrotic reaction after glaucoma filtration surgery (GFS). Therefore, the use of meldonium as eye drops in a trans-epithelial formulation with nanomicelles might represent a novel conservative treatment after GFS, prolonging the hypotonizing effects of the surgical intervention and adding a further relevant hypotonizing effect, with minimal risk of doping effects in treated patients.
Author Response
Thank you so much for your comments. We appreciate your kind remarks.